# Disclosure of Mental Health Problems or Suicidality at Work: A Systematic Review

**DOI:** 10.3390/ijerph20085548

**Published:** 2023-04-17

**Authors:** Martina O. McGrath, Karolina Krysinska, Nicola J. Reavley, Karl Andriessen, Jane Pirkis

**Affiliations:** Centre for Mental Health, Melbourne School of Population and Global Health, The University of Melbourne, Parkville, VIC 3010, Australia; karolina.krysinska@unimelb.edu.au (K.K.); nreavley@unimelb.edu.au (N.J.R.); karl.andriessen@unimelb.edu.au (K.A.); j.pirkis@unimelb.edu.au (J.P.)

**Keywords:** lived experience, disclosure, workplace, suicidal ideation, suicide distress, suicide attempt, suicidal thoughts, suicidal behaviours, suicidality, mental health, mental illness, mental ill health, stigma, discrimination

## Abstract

Many adults experience mental health problems or suicidality. Mental health and suicidality are associated with stigma and discrimination. Little is known about disclosure of mental health or suicidality problems in workplaces and the role of stigma and discrimination in affecting disclosure. To address this gap, we conducted a systematic review following the PRISMA guidelines. Searches for peer-reviewed articles in MedLINE, CINAHL, Embase and PsycINFO identified 26 studies, including sixteen qualitative, seven quantitative and three mixed-methods studies. No studies were excluded based on quality assessment. All studies reported on mental health disclosure; none reported on disclosure of suicidal thoughts or behaviours. The narrative synthesis identified four overarching themes relating to disclosure of mental health problems in workplaces. Themes included beliefs about stigma and discrimination, workplace factors (including supports and accommodation), identity factors (including professional and personal identity, gender and intersectionality) and factors relating to the disclosure process (including timing and recipients), with all influencing disclosure decision making. Significantly, this review found that there is a gap in the existing literature relating to suicidality disclosure in workplaces, with none of the included studies investigating disclosure of suicidal thoughts and behaviours.

## 1. Introduction 

The World Health Organisation (WHO) describes mental health and mental well-being as part of an individual’s overall health and states that good mental health is a fundamental human right [1]. However, not everyone experiences good mental health. As reported by the WHO, one in eight people live with a mental health condition, and mental health disorders are the leading cause of disability worldwide [1]. Closely related to mental health, suicide prevention is a global health issue, with an estimated 700,000 people dying by suicide. The WHO also reports that for every death by suicide, there are more than 20 suicide attempts [2]. Risk factors for suicidality include a complex interaction of biopsychosocial, cultural and environmental factors, which can include mental health problems [3]. As described by Pirkis and colleagues, mental health problems, which fit within the clinical and medical health domains, are part of a bigger and more complex interplay of possible, and often co-occurring, societal- and individual-level risk factors [4]. 

Stigma can negatively affect individuals living with mental health or suicidality problems. A person who has a lived experience of suicide is defined as someone who has experienced suicidal thoughts, survived a suicide attempt, cared for someone through a suicidal crisis or is bereaved by suicide [5]. Mental health falls on a continuum ranging from excellent mental health to severe symptoms of poor mental health. In this paper, we use the term “mental health problems” to refer to situations in which people experience varying issues of mental health but may not have a diagnosis of mental illness. By using mental health problems, we acknowledge that everyone experiences some issues with regard to mental health, but not everyone identifies with or has a diagnosed mental health illness, disorder or condition [6,7].

Stigma involves a process of labelling, stereotyping and separating people into more or less favourable groups and categorizing entities of ‘them and us’, which can lead to a loss of status and discrimination [8]. Self-stigma occurs when negative external stereotypes are internalised [9]. People who experience mental health problems or suicidality and are impacted by stigma often navigate complex decision-making processes when considering if, to whom, when, where and even why to disclose their mental health or suicidality problems. 

Disclosure is the sharing of personal information, from one person to another [10]. There are many reasons why a person with a lived experience of mental health or suicidality may choose to disclose or not, or even selectively or partially disclose. In providing a model for the disclosure decision-making process, Chaudoir and Fisher state that disclosure of personal, concealable stigmatised identity is motivated by wanting to achieve or maintain a sense of human connectedness [10]. In a recent literature review, the disclosure decision-making model is further described as consisting of two distinct but closely interrelated stages: pre- and post-disclosure outcomes. Hastuti and Timmins describe the pre-disclosure stage as individuals considering both internal motivating factors and external environmental factors. For the post-disclosure stage, individuals assess the potential for experiencing positive or negative outcomes as a result of disclosing. Importantly, these researchers highlight that the disclosure process is complex and operates continuously and is not simply a binary yes or no decision [11]. Disclosure risks often include the loss of status, identity or social connectedness. For these reasons, individuals may choose to delay, partially disclose or simply avoid making a disclosure. The disclosure phenomenon can also be understood by examining drivers for disclosure or non-disclosure and considering the role of relationships influencing disclosure decisions [12]. Other reviews have also contributed to what is known about disclosure in relation to mental health, stigma and discrimination [7].

As has been reported, there are often complex risks that need to be considered relating to workplace disclosure, including negative professional, social and individual-level outcomes [13,14]. However, despite the risks of the workplace disclosure of mental health or suicidality, there are many reasons why individuals may need or want to disclose at work, as disclosure benefits can outweigh negative outcomes. The potential benefits of disclosure for individuals and workplaces, as reported in recent empirical studies, include individuals receiving improved social supports and being able to access supports and accommodations, as well as improved workplace culture and acceptance towards people living with mental health or suicidality problems [14,15].

With just over half of the world’s population employed and 15% of working-age adults experiencing a mental health problem, workplace mental health and suicide prevention is important for individuals, workplaces, society and the global economy [16]. Ensuring that workplaces protect and promote mental health is increasingly seen as a public policy priority. For example, in the United Nation’s (UN) Sustainable Development Goals, goals three and eight relate to promoting health and wellbeing, employment rights and economic growth [17]. The WHO and the International Labour Organization’s policies provide a strategic framework for enhancing, protecting, and promoting good mental health, including enhancing support for workers who disclose mental health or suicidality issues. The framework recommends that workplaces develop, implement or improve mental health policies, awareness and literacy, including training for managers and all staff and improvement of supports and accommodations for workers experiencing mental health or suicidality issues [16].

Whilst people bereaved by suicide are not the focus of our review, a recent study found that participants were reluctant to disclose such issues at work, fearing negative outcomes [18]. Evidence also exists that helps to understand factors relating to a worker’s role, including workers living with mental health problems [19]. However, less is known about mental health or suicidality disclosure at work in relation to the role of stigma and discrimination in affecting disclosure decisions and managing mental ill health or suicidality at work.

Working helps promote a sense of purpose and meaning [20]. Workplaces are social structures; they provide important avenues for social connectedness and are intrinsic to self-efficacy and belonging [21,22]. Workplaces that create a safe, inclusive and equitable work cultures that support all workers, including those living with mental health or suicidality–related problems, do not only benefit employees; they also result in financial benefits for the companies themselves and the economy in general [16]. One way that workplaces could create opportunities for workers is to incorporate their lived experience insights and expertise at work. When workers are able to contribute their lived experience in meaningful ways, it can result in good outcomes for individuals, co-workers and workplaces. One example of workplace utilization of lived experience is workplace peer-to-peer support. Whilst hardly a new intervention, peer support has a proven record of achieving successful outcomes across a range of settings, including public health and emergency services [23]. However, for peer support to be mutually beneficial, sustainable and effective, individuals providing workplace-based peer support also need to be supported [24]. 

This systematic review examines mental health and suicidality problems and disclosure at work. The review aims to contribute to the existing evidence relating to identifying the role of stigma or discrimination as factors relating to workplace disclosure decision-making for workers living with mental health or suicidality problems. The review was also designed to highlight opportunities for improvement of workplace responses to disclosure, including supports and accommodations for workers living with mental health or suicidality problems. 

## 2. Materials and Methods

### 2.1. Search Strategy

The review adhered to the PRISMA 2020 statement and was registered in PROSPERO (CRD42022321179) [25]. To report on results and findings across studies that are heterogeneous, a mixed-method integrated narrative synthesis approach was used [26,27]. 

The review involved systematic searches in MedLINE, PsycINFO, Embase (all accessed via Ovid) and CINAHL. The search in MedLINE included a combination of MeSH and text words: (suicid* attempt*/OR suicid* ideation OR suicid* thought* OR suicid* behavio* OR self-harm* OR suicid*.mp OR mental health OR mental health problem* OR mental health diagnosis OR mental health disorder* OR mental health condition* OR mental health illness OR mental disorder* OR mental illness.mp. OR mental adj3 health.mp) AND (work*/OR employee* OR employer*.mp) AND (disclos*/OR nondisclos* OR non-disclos* OR self-disclos*.mp OR reveal* OR share OR sharing OR conceal* OR hide.mp) AND (stigma*/OR self-stigma* OR shame* OR discrim*.mp). Similar searches were conducted in the three additional databases. M.O.M. piloted the search strategy across the four databases, selected after consulting with a university librarian. M.O.M. conducted forward and backward citation searches of the included studies using Google Scholar. 

### 2.2. Inclusion and Exclusion Criteria

Studies were included if they (i) reported empirical data focusing on the prevalence of mental health or suicidality problems and disclosure in any workplace, (ii) used qualitative, quantitative or mixed methods, (iii) were published in English peer-reviewed journals, (iv) involved samples that included working adults aged 18 and over and (v) involved samples where at least >50% of participants were not working in lived experience designated roles, e.g., peer workers (given that individuals working in designated lived experience roles are by virtue are already ‘out’ at work, as part of fulfilling their duties). Studies were excluded if (i) they did not report empirical data (e.g., case studies, conference abstracts, commentaries, and editorials) or (ii) they were not published in English. 

### 2.3. Data Extraction

Endnote 9 was used to manage the data, remove duplicates, and facilitate collaboration among the research team. Independently, M.O.M. and K.K. screened a random 10% of the retrieved data and met to compare results. Disagreements mostly centred on interpretations of the inclusion and exclusion criteria. Discussions helped to resolve disagreements and to refine the search strategy and inclusion and exclusion criteria. M.O.M. and K.K. also independently conducted title and abstract screening, full-text screening and data extraction. Figure 1 shows the search and selection strategy.

### 2.4. Quality Assessment

M.O.M. and K.K. independently conducted the methodological quality assessment using the MMAT Mixed Methods Appraisal Tool, with disagreements resolved through discussion [28]. The MMAT Tool includes two preliminary screening questions, followed by five criteria questions, tailored to investigate methodological quality, according to study type. Criteria were assessed by recording a yes or no to each criteria question, but with no numerical ratings applied. 

## 3. Results

### 3.1. Study Characteristics

The review identified 26 studies, including 16 qualitative [29,30,31,32,33,34,35,36,37,38,39,40,41,42,43,44], 7 quantitative [45,46,47,48,49,50,51] and 3 mixed-methods studies [52,53,54]. Table 1 summarizes the included studies. All 26 selected studies focused on mental health disclosure; none sought to understand suicidality disclosure. Most of the studies were conducted in Western countries, including the United States [31,40,52,53,54], Australia [33,34,41,45,48], the United Kingdom [32,44,46,50,51], Canada [42,43,47,49], the Netherlands [30,35], Denmark [29], New Zealand [36,37] and Germany [39]. One study was conducted in India [38].

Qualitative studies collected data by conducting semi-structured interviews [31,32,33,34,35,36,37,38,40,42,43,44] and focus groups [29,30,39,41]. Quantitative studies used survey instruments to collect data [45,46,47,48,49,50,51] and mixed-methods studies collected data using focus groups, semi-structured interviews and surveys [52,53,54]. 

The qualitative studies’ methods of analysis included thematic analysis [30,31,34,36,37,40,41,42,44], content analysis [29,35,38,39,43] and discourse analysis [32,33]. Quantitative studies included descriptive analysis [46], chi-square testing [45,47,50,51], multi-level modelling and *t*-tests [48], psychometric analysis [49,50] and Fishers’ Exact tests [50]. Mixed-methods studies included descriptive and explanatory analysis and thematic analysis [52], inductive thematic analysis and confirmation factor analysis, descriptive statistics and reliability estimates to examine the quantitative data [53], and thematic analysis and statistical methods to analyse quantitative data, including *t*-tests, Pearson’s correlation and one-way analysis of variance (ANOVA) [54].

Six studies did not report on gender [32,36,45,49,51,52], fourteen included >50% females [30,31,33,34,35,37,42,43,44,46,47,50,53,54], six included either a majority of or all male participants [29,38,39,40,41,48] and two reported on transgender or non-binary participants [34,54]. Nine studies did not report on age [30,32,33,36,41,45,49,51,52]. Where reported, age ranges varied across samples, between 18 and 73 years, with a mean age of 44.9. 

Over 50% of the included studies investigated factors affecting the disclosure decision-making process [29,30,34,36,39,41,42,43,44,45,46,50,53,54]. Other studies focused on how workers managed their mental health at work [31,32,33,35,40], views about competency functioning [52], perceptions of colleagues with a lived experience, barriers to help-seeking, self-reporting and colleague-to-colleague stigma [47,48,49,51], access to accommodations [38], and maintaining open employment [37]. Only one study focused on understanding stigma and disclosure impacts [49].

Half of the studies were conducted in health settings, including in mental health settings [31,32,33,38,44,45,46,51,54]. In these studies, workers’ roles included mental health professionals, mental health nurses, psychologists, psychiatrists, peer workers, managers, case managers, generalists, doctors, nurses, health professionals, administrators, managers, and unspecified worker roles. Studies also included transportation workers [41], first responders [41,48], military personnel [29,39], and those employed in acting, nursing, butchery [40] and education [35,54]. Eight studies included workers from general populations but did not report data on workers’ roles [30,36,37,40,43,47,53,54]. Table 1 summarizes the included studies, according to study design type.

**Table 1 ijerph-20-05548-t001:** Summary of included studies.

Author (Year), Location	Eligibility Criteria	Sample Characteristics	Study Design	Main Findings
Qualitative Studies				
Bogaers et al. (2021), Denmark [29]	Soldiers with mental health or substance use conditions; soldiers without mental health or substance use conditions; mental health professionals	Participants: *n* = 46 *n* = 20 soldiers with mental health or substance use conditions; *n* = 10 soldiers without mental health or substance use conditions; *n* = 16 mental health professionals Gender: male (*n* = 37) Age: 22–57	Focus groups Content analysis	Five barriers and three enablers Barriers: fear of career consequences, fear of social rejection, lack of leadership support, lack of communication skills surrounding mental health or substance use conditions, masculine workplace cultures Enablers: anticipated positive results, leadership support, work-related mental health or substance made it easier to disclose
Brouwers et al. (2020), Netherlands [30]	People with mental illness, work reintegration professionals, employers, HR managers	Participants: *n* = 27 Gender: female 48.1%, male 37.4 % Age: not reported	Focus groups Thematic content analysis	Five themes: Being exposed to people with mental health problems helped changes attitudes Perceived stigma about mental illness from co-workers Hypothetical issues affecting disclosure decisions included having trust and support from managers Possibility of losing employment and changing attitudes of colleagues after disclosure Most workplaces/managers supportive after disclosure One unique theme (for non-mental health professionals)—mental illness is not talked about
Elliot et al. (2020), USA [31]	Mental health professionals living with mental illness	Participants: *n* = 12 psychotherapists Age: 36–63 Gender: female 75%, male 25%	Semi-structured interviews Thematic analysis	Indirect prejudice more common than indirect discrimination for 75% of participants 66% selectively shared information about their mental illness 92% believed having a mental illness an asset at work 55% described benefits to having a mental illness but said it could be a liability, interfering with job performance
Elraz (2018), UK [32]	Individuals with mental health concerns, health professionals, HR professionals, line managers, employees	Participants: *n* = 16 workers with mental health conditions Age: not reported Gender: not reported	Semi-structured interviews Discourse analysis	Negative societal representations of mental health problems translate into workplaces By gaining special skills as a consequence of their MHC, they are better positioned in their employment, compared to other colleagues Those who disclose, see themselves as champions/pioneers
Joyce et al. (2009), Australia [33]	Mental health or generalist nurses with mental health concerns (diagnosed by a medical practitioner), currently receiving treatment for mental ill health and not in a state of acute active psychosis	Participants: *n* = 29 mental health nurses Age: 24–56 Gender: female 82.7%, male 17.2%	In-depth interviews Discourse analysis	Four subthemes: Declaring mental illness and decision to disclose to co-workers and nurse managers and implications of disclosure for identity as a confident and capable nurse, not somebody who needed therapeutic interventions Collegial support—some management and colleagues will be deliberately destructive to the wellbeing of nurses Some managers are caring, sensitive and supportive Enhancing support—nurses, as well as the nursing profession, has a responsibility to develop an awareness of their attitudes towards mental health
King et al. (2021), Australia [34]	Mental health professionals identifying with lived experience (publicly or privately), mental health professionals not identifying with lived experience, staff in designated lived experience roles (peer workers), staff in supervisory roles to the above groups	Participants: *n* = 33 mental health workers Age: not reported Gender: females 78.7%, males 15.1%, non-binary 6.0%	Semi-structured interview Thematic analysis	Three findings Peer workers reveal the impact of organizational differences in supporting sharing Importance of identity and identity management at work Influence of team culture affects how sharing occurs
Lyhne et al. (2021), Netherlands [35]	(a) Higher educational level, minimum Master’s degree (b) Diagnosed with depression by a general practitioner, psychiatrist or medical specialists in psychiatry (c) Employed while diagnosed with depression (during the last 18 months) (d) Danish or English speaking	Participants: *n* = 8 Age: 35–66 Gender: female 62.5%, males 37.5%	Semi-structured interviews Content analysis	Four categories (opportunities/challenges) Struggling with acknowledging depression and disclosure Fear of stigma—disclosure avoided fearing stigma from co-workers/employer Work is a life motivator—work contributes with meaning and substance in life, provides opportunities for managing work participation Striving to complete work tasks at the expense of private life; a strong work identity and commitment to work impose high self-expectations as a worker
Peterson et al. (2011), New Zealand [36]	Employed and self-identified as experiencing mental ill health	Participants: *n* = 22 Age Range: not reported Gender: not reported	Semi-structured interviews Thematic analysis	Pressure to not disclose was due to fear of discrimination Pressures to disclose were legal, practical and moral Legal pressures related to health-related questions on job application forms Practical pressures included ensuring employers could allow time off for appointments or other accommodations Moral pressures included disclosure was the “right thing to do”
Peterson et al. (2017), New Zealand [37]	(a) Employees (mental health service users, in open unsupported employment) if employer also agreed to be interviewed (b) Employers (if approached by employee)	Participants: *n* = 30: *n* = 15 employees *n* = 15 employers Age: employees (*n* = 5, 23–34), (*n* = 8, 45–54), (*n* = 1, 55–65) Gender: female 60%, male 36.6%	Semi-structured interviews Thematic content analysis	Critical factors include work has meaning Disclosure mostly occurs at start of employment There are benefits of working Special arrangements, accommodations and flexibility are important The working environment is one of the main reasons employees enjoyed work
Rangarajan et al. (2020), India [38]	Clinically stable people with serious mental ill health with a score of ≤2 on the Clinical Global Impression (CGI) Severity scale, currently employed or employed in the past for at least six months and able to provide valid interviews; Mental health professionals having >15 years of formal education); Employers who had employed people with mental health concerns	Participants: *n* = 15 employees (*n* = 5); mental health professionals (*n* = 5); employers (*n* = 5) Age: M (SD) 38.26 (8.94) years Gender: females 20%, males 80%	In-depth interviews Content analysis	Reasonable accommodations and supports improves work efficiency, modifications in the workplace environment, modifications in the appraisal, integration of mental health and employment services, and supportive employer policies Undue burden a major concern for employers and mental health professionals, compared to those with lived experience Stigma reduction and disclosure of mental health problems are important prerequisites for reasonable accommodations
Rusch et al. (2017), Germany [39]	(a) Active-duty soldiers receiving psychiatric inpatient treatment; (b) Soldiers not diagnosed with mental illness	Participants: *n* = 56 Age: M (SD) 34.8 (9.4) Gender: male 91%	Focus group Content analysis	Soldiers with serious mental health ill health struggle with disclosure decisions Disclosure remains deeply personal and difficult; decisions are shaped by circumstances, some beyond the individual’s control, such as the stigma associated with mental ill health in society and in the military Stigma remains a barrier to reintegration and recovery
Siegel et al. (2020), USA [40]	Working men who reported a clinical diagnosis of an eating issue	Participants: *n* = 14 Age: (M = 27.86; SD = 6.95) Gender: male 100%	Semi-structured interviews Thematic analysis	Fear of stigma and (non) disclosure, emotional reactions, coping strategies, and impaired work performance Vigilance required to remain undetected combined with the pressure to present with masculinity at work made work life challenging White men may be negatively impacted by their social locations
Stratton et al. (2018), Australia [41]	(a) Transport or first responder workers with mental health concerns who had disclosed; or (b) occupying a position of authority, i.e., supervisors (whose employees with mental health concerns have disclosed to them)	Participants: *n* = 13 Age: not reported Gender: female 38.4%, male 61.5%	Focus groups Thematic analysis	Male-dominated workplace affected disclosure decisions, even for those who had already done so, as influenced by barriers to disclosure Six negative themes of internal and external factors influencing the decision-making process: knowledge about symptoms, self-discrimination (internal), stigma and discrimination by others, limited managerial support, dissatisfaction with services, and/or a risk of job or financial loss (external)
Toth et al. (2014), Canada [42]	Post-secondary education employees diagnosed with a mental disorder recognized under the DSM-IV-TR (3) by a qualified health professional	Participants: *n* = 13 Age: 21 to 55 Gender: female 76.9%, male 23.0%	Semi-structured interviews Thematic analysis	Default position of nondisclosure adopted due to fear of stigma Employees need a reason to disclose Decision-making process involves a risk–benefit analysis Training managers and staff would help reduce stigma Training managers to address power imbalances should be implemented
Toth et al. (2021), Canada [43]	>18 years of age, primary diagnosis of mental health problems (self-reported), obtained competitive employment within the last 12 weeks and able to communicate in English or French	Participants: *n* = 28 Age: M = 37.79, SD = 11.29 Age range: 22–59 Gender: female 57.1%, male 42.8%	Semi-structured interviews Thematic content analysis	Goals and conditions/context important antecedents for disclosure decisions Participants reported a psychological and physical release after disclosure, described as decrease of pressure/cognitive load Majority who disclosed perceived receiving a positive response from their supervisor, provided hope Fear of negative responses confirms there is still work to be done
Waugh et al. (2017), UK [44]	NHS trust employees, both mental health and general health service employees	Participants: *n* = 24 Age: 18–60 Gender: females 66.6%, males 33.3%	Semi-structured interviews Thematic analysis	Personal experiences with people with mental health problems helped change attitudes Perceived stigmatising views of mental ill health in other staff members Hypothetical factors affected disclosure decisions Attitudes towards disclosure—risks and concerns for potential of discrimination or stigma Support after disclosure—managers generally described as helpful and supportive One unique factor of non-mental health professionals was mental ill health not talked about at work
Quantitative Studies				
Brennan et al. (2019), Australia [45]	Health professionals with long-term conditions	Participants: (*n* = 614), subset analysis (*n* = 545) Age: not reported Gender: not reported	Survey frequency, descriptive and inferential statistical analysis	Self-disclosure decisions are multifactorial: age, gender, workplace circumstances and nature of health condition Medical professionals less likely than nurses and allied health workers to disclose to colleagues People with mental health problems more cautious and selective in disclosing and more likely to disclose to supervisors than to colleagues
Cohen et al. (2016), UK [46]	Qualified doctors, regardless of whether or not they had experienced mental ill health	Participants: *n* = 1954 Age: 18–>65 Gender: female 60%	Survey Cross-sectional analysis	Younger doctors less likely to disclose than general practitioners and consultants Concerns about being labelled, confidentiality and not understanding the support structures available identified as obstacles to disclosure Doctors likely to disclose mental health problems later than they anticipated
Dewa et al. (2014), Canada [47]	>18 years, living in Ontario with workforce participation during 12 months prior	Participants: *n* = 2219 Age: <30, 30–39, 40–49, 50–59, 60–64, and >65 Gender: female 63.8%, male 36.0%	Survey or telephone questionnaire Patient Health Questionnaire-8 (PHQ-8), X2test	Although critical for workers who experience a mental health problem and find work challenging, a significant proportion do not seek support Fear of negative repercussions a barrier to disclosure One third would not tell managers if they experienced mental health problems 50% of workers identified that improving policies and practices would encourage disclosure 35% identified a combination of factors as important to encouraging disclosure: workplace relationships, supportive colleagues and good practices and policies Most pervasive reasons for concerns about a colleague with a mental health problem included safety and colleague’s reliability
Marshall et al. (2021), Australia [48]	Police officers who participated in a mental health screening program (and aware that individuals reporting significant levels of symptoms may be offered a follow-up assessment and counselling)	Participants: *n* = 90 Age: M age 44.1 years (SD = 9.3) Gender: female 31.1%, male 68.8%	Cross-sectional survey Depression Anxiety Stress Scales (DASS-21) for symptoms of psychological distress, abbreviated version of the PTSD Checklist for DSM-5 (PCL-5) for symptoms of post-traumatic stress disorder (PTSD), paired sample *t*-tests, independent sample *t*-tests	Employees under-reported symptoms when completing screening administered by their employer Under-reporting occurred regardless of gender and symptom type Senior staff and those with the most severe post-traumatic stress disorder and common mental health symptoms more likely to under-report Employer-administered mental health screening is not able to accurately capture all mental health problems
Stuart, H. (2017), Canada [49]	Front-line police officers who attended a one-day mental health workshop	Participants: *n* = 133 Age: not reported Gender: not reported	Pre-workshop surveys and post-workshop validity checking Psychometric analysis using 12-item Perceived Devaluation and Discrimination scale	Police-to-police mental health stigma may be a strong feature of police culture Police should be a focus for targeted anti-stigma interventions Police Office Stigma Scale may provide important insights into the nature and functioning of police-to-police stigma in police cultures in future research
Tay et al. (2018), UK [50]	Qualified clinical psychologists living in the UK	Participants: *n* = 678 Age: 84.2% 30–59 Gender: female 82.1%, male 17.8%	Survey The Social Distance Scale (SDS), nine-item Stig-9, 10-item self-stigma subscale of the Military Stigma Scale (MSS), nine-item Secrecy Scale, 10-item Attitudes towards Seeking Professional Psychological Help Scale—Short Form, Chi-square and Fisher’s exact tests, One-way ANOVA and independent samples *t*-tests	Two-thirds had experienced mental health problems Perceived mental health stigma higher than external and self-stigma Participants more likely to have disclosed in their social domains than at work Negative consequences for self and career and shame prevented some from disclosing and seeking help
White et al. (2018), UK [51]	Psychiatrists working in the West Midlands region	Participants: *n* = 370 Age: not reported Gender: not reported	Questionnaire X2 tests	Most reluctant to disclose to colleagues or professional organisations Choices regarding disclosure and treatment influenced by confidentiality concerns 66%, stigma 22%, and career implications 35%, rather than quality of care 16%
Mixed Methods Studies				
Boyd et al. (2016), USA [52]	Mental health professionals with mental health problems, employed by Department of Veterans Affairs (VA), in Veterans Health Administration (VHA), who had been meeting monthly	Participants: quantitative (*n* = 77) qualitative (*n* = 55) Age: not reported Gender: not reported	Survey Descriptive statistics and manual descriptive coding for qualitative data	Very few asked for accommodations Two-thirds had not disclosed to their patients Respondents disclosed to only 16% of their colleagues, and about one third had not disclosed to any colleagues Lived experience was an asset, whether or not disclosed Many are proud to stand up and be counted, others cited reasons to be cautious about disclosure
Follmer et al. (2021), USA [53]	18 years of age or older, employed at least 20 h per week, previous formal diagnosis of depression	Participants: Study 1—*n* = 30 Study 2—*n* = 455 Study 3—*n* = 233 Age: Study 1: average age: 34 years; age range: (19–65 years) Study 2: not reported Study 3: not reported Gender: Study 1: female 73%, male 27% Study 2: not reported Study 3: female 73%, male 27%	In-depth phone interviews, survey Thematic analysis, approach and avoidance scales, multivariate analysis	Approach and avoidance motives influenced by multiple factors, including social support, stigma, and diversity climate MANCOVA results not significant, λ = 0.99, F(2, 219) = 1.37, *p* > 0.05, η2 = 0.01 and no significant differences in the reported means for engagement as a function of the decision to disclose (M = 4.26) or conceal (M = 4.29), presenteeism did not significantly varied as a function of disclosure (M = 2.91) or concealment (M = 2.73)
Marino et al. (2016), USA [54]	Individuals with lived experience	Participants: *n* = 117 qualitative: *n* = 35 quantitative *n* = 117 Age: Qualitative: M 47, SD 10.86, range 25–71 Quantitative: M 47, SD 12.27, range 21–71 Gender: Qualitative: female 52.4%, male 42.8%, transgender 2.8% Quantitative: female 70%, male 26.8%, transgender 0.2%	Semi-structured interviews, focus groups (in-person or by phone) and survey Analysis: Independent t tests, Pearson correlation, one-way analysis of variance (ANOVA)	Lived experience a resource to assist others with service delivery Lived experience foundational to building relationships with individuals in recovery Disclosure dependent on social context and perceptions of safety and power differentials Individuals concerned regarding exclusion and discrimination

M: Mean; SD: Standard Deviation.

### 3.2. Quality Assessment

Using the MMAT tool, no studies were excluded based on study quality. The interrater reliability was moderate (κ = 0.59) [28]. The three mixed-methods studies were all deemed to have good quality [52,53,54]. However, due to missing information, including poorly defined data analysis methods, three of the qualitative studies resulted in ratings of ‘can’t tell’ relating to vaguely defined data analysis methods [32,33,34]. Only one of the quantitative studies was deemed to have good quality according to the MMAT criterion [51]. Quantitative studies’ methodological issues related to missing recruitment and selection strategy information [49], non-representative of target population sampling techniques [47,48,49] and possible risk of bias due to non-responses to survey instruments [45,46,47,48,50]. A summary table, inclusive of comments, is included in Appendix A. 

### 3.3. Main Themes across All Studies

As depicted below in Table 2, beliefs about stigma and discrimination played a significant role in informing and influencing disclosure decisions. Across all studies, workplace factors also played a significant role. These workplace factors included workplace culture, policies, procedures, training and supports, and accommodations. Closely related to these factors, many of the studies had findings relating to the importance of the disclosure process (including timing and recipients) as key to informing disclosure decisions. Other factors affecting disclosure and non-disclosure related to identity, in relation to personal and professional identity, intersectionality and, in some instances, the role of gender and societal norms informing disclosure decisions. A narrative synthesis, according to study design, is also presented below. 

### 3.4. Qualitative Studies

#### 3.4.1. Beliefs about Stigma and Discrimination

Stigma, including self-stigma, has been widely reported relating to informing disclosure decisions. Two studies involving soldiers with and without serious mental health or substance use problems reported that stigma, self-stigma and fear of discrimination informed disclosure decisions. However, these participants also reported that once a disclosure had been made or when a mental health problem could no longer remain hidden, disclosure supported help-seeking behaviours and recovery [29,39]. Fear of losing employment, negative career impacts and competency concerns were reported to have informed disclosure and non-disclosure decisions [29,30,33,40,41]. Workplace stigma and a lack of mental health literacy also seemed to affect how mental health was perceived, which affected workers’ disclosure decisions [29,32,39]. Workers also reported experiencing direct and indirect discrimination from management and colleagues [31,33,36,44]. Having previous negative experiences of disclosure and workplace culture also appeared to affect disclosure choices [29,35,39,42,44]. Higher levels of self-stigma affected disclosure, particularly for men working in male-dominated settings [29,39,41]. 

#### 3.4.2. Workplace Factors, Supports and Accommodations

Studies identified that workplace factors, including supports and accommodations, affected, or at least informed, disclosure and non-disclosure decisions in both positive and negative ways. Some workers reported receiving good support and empathy after disclosing [29,30,33,43,44]. Some participants felt that workplaces had an obligation to provide support and accommodations and indicated that this was a key factor in their decision to disclose [33,36,37,41]. Another motivating factor was that the benefits of employment could facilitate disclosure. Work was seen to provide meaning and purpose, which in turn helped workers manage their mental health at work while contributing to society [35,37]. Redesigning workplace policies was also found to be key to helping improve conditions for workers living with mental health problems [37,38,39]. Providing training to managers, co-workers and employers was also reported as important to improving workplace disclosure. Participants reported that providing training would help improve mental health awareness and stigma reduction, help to educate employers about their responsibilities relating to providing support and accommodations and would assist managers to be better equipped to respond to and mitigate any issues relating to worker–manager power imbalances [38,42]. 

#### 3.4.3. Identity Factors

Personal and concealable identity as well as managing one’s professional identity were reported as significant factors in disclosure and non-disclosure decisions. Workers reported feeling conflicted, wanting to be seen as capable and responsible, irrespective of their mental health problems [32,33,34]. Closely related to these identity conflict issues, perfectionism and unrealistically high self-expectations were reported as challenges to maintaining a professional identity alongside managing one’s personal mental health [35]. Four studies identified identity and disclosure factors for men, including men feeling burdened by the weight of gender, and societal norms overlayed with the experience of living with poor mental health at work [29,39,40,41]. 

#### 3.4.4. Disclosure Process Factors: Timing and Targets for Disclosure

Qualitative studies found that the disclosure decision-making process was complex and included multiple factors, leading to sophisticated disclosure strategies such as signalling behaviours, selective disclosures, preferring to disclose to supervisors (for reasons of trust) and choosing to disclose when information could be voluntarily and confidentially shared [29,30,31,33,34,42,44]. In terms of the timing of disclosure, some inconsistencies were noted. For example, one study involving workers who were also mental health service users found that these participants preferred to disclose at the beginning of employment and during the hiring process [37]. However, in another study examining attitudes towards the timing of disclosure for three groups of participants, including people with lived experience, HR managers and return-to-work specialists, there was little agreement among the groups about when was the ‘right time’ to disclose. HR managers and return-to-work specialists preferred disclosure to occur at the beginning of employment, for example, during the hiring process. This view was not supported by the workers with lived experience [30]. Another factor key to understanding the disclosure process related to workers living with mental health problems feeling that they could gain the support of co-workers and managers with whom to trust with disclosure [43]. Some participants also reported that having co-workers who had prior exposure to people with mental health challenges helped pave the way towards ensuring safe and supported disclosures [44]. 

### 3.5. Quantitative Studies

#### 3.5.1. Beliefs about Stigma and Discrimination

In a study focusing on patterns of self-disclosure for people with a variety of health conditions including mental health, those living with mental health problems were less likely to disclose, compared to people living with physical health problems [45]. Fears about negative career consequences were reported as barriers to safe disclosure by one third of participants in a Canadian general population study [47]. In this study, 64% of participants without mental health problems indicated they had safety concerns about colleagues with mental health problems, with men more concerned than women and managers more concerned than co-workers. However, 49.5% of these same participants reported that despite their concerns, they would offer support to people experiencing mental health problems [47]. Another study found that 85% of police officers would avoid disclosure; fearing potential negative consequences of discrimination was reported by 62% of these police officers [49]. This study also found that police-to-police stigma was high. Combined, these factors served as barriers to safe disclosure [49]. Perceived stigma was found to be a more significant factor than external or self-stigma in a study involving clinical psychologists, suggesting that the potential threat of stigma may limit disclosure and help-seeking [50]. In another study involving psychiatrists, stigma, fear of negative career consequences and potential loss of professional standing was reported, with 34.7% fearing career impact, 22.4% concerned about stigma, and 22.4% indicating these concerns served as barriers to disclosure [51]. 

#### 3.5.2. Workplace Factors, Supports and Accommodations

Some studies identified that workplace factors helped shape disclosure decisions. In a study focused on understanding attitudes toward disclosure, supportive managers, good workplace policies and training were found to increase the likelihood of disclosure [47]. However, this study also noted that because participants felt these systems were not yet in place, they were reluctant to disclose. The importance of workplace training, specifically anti-stigma training, was also identified in a study involving police officers [49]. 

#### 3.5.3. Identity Factors

Identity and identity management, including navigating personal and professional identity conflicts, was found in studies conducted within the health sector. Unsupportive workplace cultures, combined with personal demographic and cultural characteristics, were found to negatively affect healthcare workers’ disclosure decisions [45]. One study, which included doctors with and without mental health problems who were asked questions about actual or hypothetical disclosure, found that 54% would not disclose. Interestingly, those with mental health problems said they would disclose later than those who were asked hypothetically about disclosure; notably, younger and trainee medical professionals stated they would be less likely to disclose than their older and more experienced counterparts [46]. A similar study involving health professionals found that doctors were less likely to disclose compared to nurses and allied health professionals [45]. Significantly, personal experiences of mental health problems amongst clinical psychologists were high, with two-thirds declaring they had experienced mental health challenges [50]. Workplace-related professional identity factors also appeared to limit disclosure amongst first responders, with one study finding that police under-reported mental health problems. This study also found that lower-ranked officers and those with more severe mental health conditions, such as post-traumatic stress disorder, would be less likely to disclose [48]. Notably, in this same study, there were no reported gendered differences relating to self-disclosure. This may point to the fact that strong police culture affects all police equally, regardless of gender, as was reported in one other selected study [48]. 

#### 3.5.4. The Disclosure Process: Timing and Targets for Disclosure

One study found the disclosure decision-making process was complex and influenced by factors such as age, gender and health concerns [45]. This study also found that participants with mental health problems were more likely to selectively disclose and to do so with a supervisor rather than to a co-worker [45,50]. Concerns relating to power dynamics were also influential in terms of the timing and targets for disclosure [50]. For some professions, such as doctors, respondents stated they would be more likely to delay disclosure [46]. 

### 3.6. Mixed-Methods Studies

#### 3.6.1. Beliefs about Stigma and Discrimination

In one study, fears related to stigma and discrimination resulted in participants avoiding disclosure [53]. In another study involving a range of mental health workers living with mental health problems, only 13% were in favour of disclosing and over a third remained cautious when considering disclosure, citing stigma concerns. [52]. In another study also involving mental health workers, disclosure decision-making factors included having an awareness of power dynamics and considering the type of role a person held due to concerns relating to possible stigma [54]. 

#### 3.6.2. Workplace Factors, Supports and Accommodations

In one study, 15% of mental health professionals (e.g., psychologists, psychiatrists, and social workers) living with poor mental health stated they would disclose to ask for support [52]. In two studies, factors such as organisational culture and supportive systems promoting workplace diversity were identified as important disclosure considerations [53,54]. 

#### 3.6.3. The Disclosure Process: Timing and Targets for Disclosure

Similar themes were identified relating to the disclosure process in all three mixed-methods studies. One study found that the disclosure decision-making process was indeed complex and dependent on social contexts and an awareness of power differentials [54]. Mental health stigma seemed to drive mental health professionals’ decisions involving timing and recipients of disclosure. Notably, in this study, less than a quarter of workers reported having disclosed to co-workers or managers [52]. Having trust in the co-worker or manager with whom the information would be shared was reported as a key factor [54]. The complex nature of the disclosure decision-making process was also highlighted in a study that found that the key to predicting whether or not the disclosure would result in a positive or negative outcome could often be traced back to a person’s motivations for electing to either disclose or conceal depression [53]. 

## 4. Discussion

This review identified 26 studies, mostly from high-income countries, which were published between 2009 [33] and 2021 [29,34,35,48,53]. The review sought to understand disclosure of mental health or suicidality problems in workplaces and to identify what, if any, role stigma and discrimination play in influencing disclosure decisions. Studies involved working-age adults working in a variety of industries and professions; over half of these studies were conducted in health and mental health settings. Ages of sample participants ranged from 18 to 73 years. The review identified a significant gap in the literature. All of the included studies addressed mental health disclosure, and none addressed disclosure of suicidality.

Overwhelmingly, stigma and discrimination, including fear of stigma, self-stigma and discrimination (including prior experiences of stigma or discrimination) appear to influence decision making around disclosure, as found in our review and reported in previous research [7,55,56]. The review findings suggest that fear of stigma may be a more significant factor than self-stigma [35,36,39,40,42,43,44,47,50,51,53]. However, there are some noteworthy gendered differences, with self-stigma and colleague-to-colleague stigma being more likely to negatively affect males, particularly men working in male-dominated industries [29,39,40,41,48,49]. Stigma and discrimination were also found to be more prevalent amongst health and mental health professionals [30,33,45,46,50,51] The review identifies a number of important factors that moderate both the threat and consequences of stigma and discrimination. These factors include being able to disclose voluntarily, at a time of one’s own choosing, and having opportunities to become a positive role model for hope and recovery, which can also help to dispel the negative public narratives that portray people with mental health problems in less than favourable and counter-productive ways [30,32,33,44,52]. 

Fostering workplace climates conducive to disclosures of mental health or suicidality problems is within reach of all workplaces [33,34,37,38,42,43,44]. These initiatives include creating workplace cultures through the development of policies and procedures that support all workers, combined with having supportive co-workers and managers, trust, privacy, and access to timely and appropriate levels of support and accommodations. However, if one or more of these important ingredients is missing (such as working in an unsupportive culture or concerns relating to competency or mandatory reporting requirements), these factors can create real challenges to disclosure and help-seeking behaviours. A workplace that is able to clearly communicate its procedures for accessing workplace supports and accommodations also seems significant. When support and accommodations are not clearly communicated or are poorly understood, this can lead to workers feeling uncertain regarding the availability of supports and accommodations, which creates further barriers to disclosure, as found in our review and in a previously published systematic review [6,39]. 

Our review also supports findings by Zamir and colleagues and Hastuti et al., whose reviews found that the potential benefits from mental health disclosure could enhance workplace cultures [6,11] Our review found that when the workplaces are supportive, including ensuring good access to supports and supportive managers, workers are more likely to voluntarily disclose, and the disclosure experience is more likely to be a positive one for all involved [29,30,33,38,43,44]. Additionally, workplaces that can provide anti-stigma and mental health awareness training to managers can help them to respond with compassion and an awareness of any power imbalance that may exist between supervisors and supervisees [42,51]. Similarly, a workplace culture that supports and encourages all workers to thrive, including those living with mental health or suicidality issues, may contribute to more workers voluntarily disclosing at work [34,37,47,53,54]. 

The review showed that identity, including managing personal and professional identity in the face of suicidality or mental health problems, is a significant factor influencing disclosure and non-disclosure choices. Identity factors are also dependent on a range of personal characteristics, such as a worker’s role, gender, societal norms, and expectations often associated with a particular type of professional role [29,34]. Two studies found that these factors were barriers to disclosure when accompanied by increasing levels of self-stigma and shame [40,41]. It also seems that identity conflicts are more prominent for particular types of workers, such as those working in helping professions, including first responders and healthcare professionals (including peer workers). These workers are required to utilise high levels of compassion and empathy at work. Yet, while providing vital empathetic human-centred care and striving to maintain high levels of professionalism, these workers seem to face even bigger hurdles when it comes to managing their own lived experience, ultimately and counter-intuitively affecting their disclosure and help-seeking behaviours at work [33,34,45,46,50,51].

Disclosing mental health or suicidality problems is fraught with challenges and exacerbated by fears relating to stigma and discrimination. One way that workplaces could help to improve responses to disclosure and provide support to individuals living with mental health or suicidality challenges would be enhancing opportunities to champion these workers [30]. By maximising the unique insights of people living with suicidality or mental health problems and creating opportunities for these individuals to take on important advocacy and recovery-focussed roles, all parties could benefit by introducing proactive and preventative peer-to-peer workplace support [7,49]. 

The road to disclosure of suicidality or mental health problems is complex, filled with competing tensions, including motivational and avoidance factors, as reported in this review and in a recently published scoping review [57]. It seems that key contributing factors to disclosure decision making are having adequate time to plan for disclosure and to strategise about desired outcomes and make informed decisions about the level of disclosure, timing, place and, importantly, to whom to disclose [30]. Participants in our review reported that engaging in signalling behaviour and selective disclosure helped to ‘test the waters’ and to see how receptive and supportive fellow co-workers and managers would likely be [29,31,34,42,45]. Understanding how the disclosure process relates to the receiver of disclosure is also closely related to this. A number of studies reported that, more often than not, this choice is multifactorial, involving trusting the recipient and believing that the response to the disclosure would elicit a positive and supportive response. Many of the studies included in our review reported preferencing disclosing to trusted supervisors as a key ingredient in the disclosure process. In most cases, these decisions were well-founded, with managers responding favourably and able to offer adequate supports or accommodations [30,33,43,44,45]. Closely related to this, having confidence and trust that information will remain private also seems to be key to ensuring the disclosure experience is a positive one. Personal autonomy and dignity, including having the right to voluntarily share or withhold personal and concealable information, also helps to ensure a positive disclosure experience and outcome. Relating to workplaces as social structures, our review also found that a key factor informing the disclosure process relates to acknowledging and mitigating any power imbalances during the disclosure event [33,42,46,54]. 

There are many current universal workplace interventions targeted at addressing workplace stigma to help meet the needs of workers living with suicidal thoughts and behaviours or mental health problems. For example, a recent review found that anti-stigma programs designed to suit various workplace contexts resulted in positive outcomes [58]. In Australia, programs such as Mental Health First Aid, Mates in Construction and Living Works Australia all offer various interventions, including gate-keeper programs, tailored to workplaces. A recent study evaluating an adapted version of the standard Mental Health First Aid program for people experiencing suicidality (Conversations for Life) found that after attending the intervention, participants reported increased levels of confidence in responding to co-workers experiencing suicidality [59]. Additionally, many workplaces also offer Employee Assistance Programs to support workers with mental health or suicidality problems. However, as found in this review, whilst EAP services may suit some workers, concerns over confidentiality mean that some workers are reluctant to access these services [41]. 

It is important to note that many of the above-mentioned interventions place the onus on individuals to manage their mental health or suicidality at work, including decision making and help-seeking relating to disclosure. Selective interventions have an important role to play in assisting workplace disclosure. These targeted interventions include decision-making tools that can help an individual worker make better-informed disclosure decisions [43,60]. However, as these two studies also report, and as substantiated by our review, in order for the disclosure to result in a positive outcome, it is often contingent on workplace factors beyond the individual’s control. These workplace factors include having a supportive workplace culture and a compassionate supervisor to whom to disclose [43,60]. As evidenced in our review, there is a need to implement universal approaches that include mental health awareness and promotion, which can address stigma and discrimination in workplaces while responding to the needs of all workers, particularly those experiencing suicidality or mental health problems. Focusing on workplace-wide system-level improvements would help to ensure that workplaces are meeting their legal and moral obligations in relation to promoting worker wellbeing and that individuals are not solely responsible for self-managing their mental health or suicidality. 

## 5. Limitations

This review has some limitations involving both the study design and selected studies’ findings. The selected studies were primarily conducted in high-income countries, with the exception of one study from India. Studies were limited to those published in English. As such, the review is limited in its ability to provide significant culturally informed findings. In addition, we only included studies whose samples included workers over the age of 18. Therefore, it is difficult to generalise our findings for all workers, including younger workers. Similarly, most studies included more females in their samples, making these findings difficult to generalise across genders and gender-diverse populations. Overall, the quality of included studies ranged from poor to good, which suggests that the findings from at least some of the studies should be interpreted with caution. Future research should seek to include research from more diverse populations and geographical regions. 

## 6. Conclusions

Despite finding a significant gap in the existing literature relating to suicidality disclosure at work, it is clear that the disclosure for working adults living with suicidality or mental health problems is complex, nuanced and involves assessing risks versus benefits, which influences disclosure and non-disclosure decisions. Many of the challenges relate to real or anticipated threats of stigma and discrimination. However, as found in this review, there are many opportunities that could help to improve the disclosure experience, workplace responses and the provision of compassionate and reasonable supports and accommodations.

Some of the key challenges for workers living with mental health or suicidality problems, who may need or want to disclose at work, relate to implementing universal changes in workplaces. These recommendations include improving workplace policies and procedures, delivering anti-stigma, mental health and suicide prevention literacy training, and fostering opportunities for worker-to-worker peer support and advocacy. Collectively, if introduced, these measures would help create vibrant and healthy workplaces that celebrate and support all workers to thrive, including those living with suicidality or mental health problems. 

Based on our findings, there is a great need to conduct research focussed on understanding disclosure for people experiencing suicidality at work. Future research should seek to investigate the workplace experiences of those living with suicidal thoughts and behaviours, including identifying barriers and opportunities for improvements, as identified by a range of stakeholders, such as workers with and without a lived experience of suicidality, their managers and employers. Future research would also help to identify appropriate and effective workplace responses, supports and accommodations for people experiencing mental health or suicidality–related challenges at work.

## Figures and Tables

**Figure 1 ijerph-20-05548-f001:**
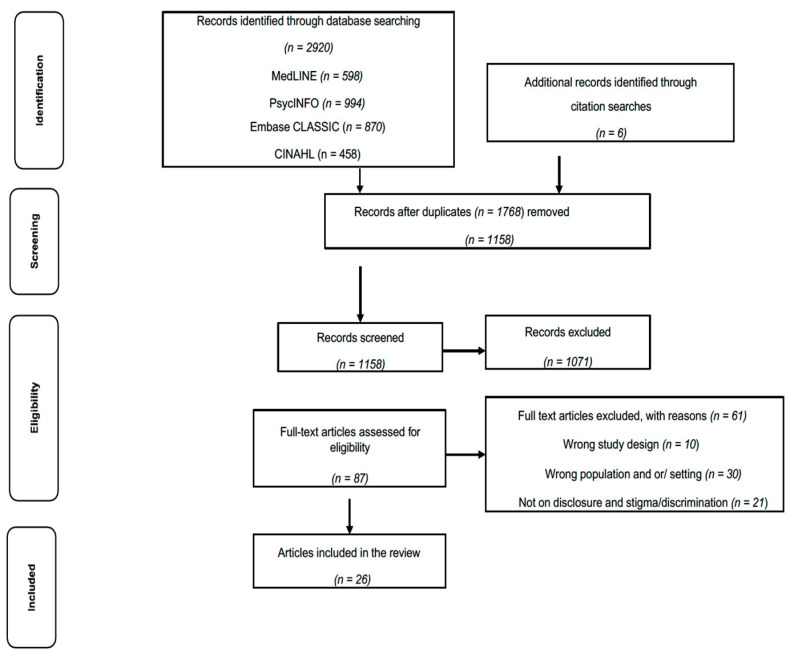
PRISMA Flow Diagram.

**Table 2 ijerph-20-05548-t002:** Summary of main themes.

Theme	Qualitative Studies	Quantitative	Mixed Methods
Beliefs about stigma and discrimination	[29,30,31,32,33,35,36,39,40,41,42,44]	[45,47,49,50,51]	[52,53,54]
Workplace factors (including supports and accommodations)	[29,30,33,35,36,37,38,39,41,42,43,44]	[47,49]	[6,52]
Identity factors (including personal and professional identity, gender and intersectionality)	[29,32,33,34,35,39,40,41]	[45,46,48,50]	
Disclosure process factors (including timing and recipients)	[29,30,31,33,34,37,42,43,44]	[45,46,50]	[52,53,54]

## Data Availability

The data presented in this review are available upon request from the corresponding author.

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
