# Peer review of "Disclosure of Mental Health Problems or Suicidality at Work: A Systematic Review"

_ijerph, 2023, doi:10.3390/ijerph20085548_

Round 1
Reviewer 1 Report
This is a systematic review of the disclosure of mental health and suicide concerns at work.
In the introduction.
The work does not determine precisely what is a mental health concern. This is a term that seems to include mental disorders, diagnoses of mental disorders, mental illness, and so on.
Line 33 refers to “Australians aged 16 - 85 experienced a mental health concern” when the study clearly mentioned mental disorders.
The methodology is clearly described.
The results are clearly described, for the most part.
In Table 1 I do not see the relevance of reporting the statistical software used. The relevant data is the statistical tests used. In this same table 1, in the citation White et al. (2018) United Kingdom [50] reports data that is not understandable ("P40.05 considered significant"). It is necessary to standardize the information mentioned in "study design" since sometimes they refer to the instruments used and on other occasions, they do not report it. In case of not having the information, it is necessary to declare it, as is done in the column of participants.
The discussion section addresses the most relevant findings.
The limitations section addresses the major drawbacks of the study.
The conclusions section opens with the absence of studies on the disclosure of suicidal thoughts and behaviors. Then discuss the findings. In my opinion, the study's positive findings should be considered first, to address the negative findings later.
In appendix A Quality assessment in point 4.2 it is not possible to know the comments to which studies they refer. It is necessary to note the citations to each comment. In point 4.4, reference 44 is not included in the comments. It is necessary to have it.
There are some typographical errors.
On lines 192 and 193.
In lines 200 and 201, it is mentioned that there are three qualitative studies, but only two are cited. It is necessary to correct it accordingly.
In line 374, “concerns” appears.
Although the study included disclosure of suicidal concerns in its review, the fact that there are no studies in this regard is a significant limitation for the manuscript to be included in the special issue.
Author Response
Dear Expert Reviewer,
We thank you for your feddback and advice.
Please see the attached file with our responses.
Kind regards,
Authors

Reviewer 2 Report
REVIEW OF IJERPH-2276583
The article presents a systematic review of extant literature on disclosing mental health concerns and suicidal thoughts at work. The paper addresses an important issue but I have several concerns mostly relating to the paper’s theoretical focus and practical implications.
· The definition of mental health should be provided.
· Please describe how mental health is related to suicidal ideation. This needs to be clarified early on since the authors begin with mental health and then suddenly focus on suicidal ideation.
· Consider describing risk factors that contribute to suicidal ideation.
· What are the prevalence rates globally, especially since this is a systematic review and not an empirical paper based on an Australia?
· Why should be suicidal ideation and/or mental health problems be disclosed in the workplace? Do the advantages overcome the disadvantages?
· Also, workplaces tend to be very busy and very impersonal, according to organizational psychology. Why should a colleague care if one has a lived experience of suicidal ideation or attempt?
· Why have the authors not considered other databases such as PubMed which also includes mental health research studies?
· Please divide the introduction using subheadings. Too much information is presented without clear structure.
· Describe the decision-making process of disclosure. What are the steps before one decides to disclose (in any setting).
· Line 84: lived experiences should be clearly defined as lived experiences of suicidal ideation, otherwise it is vague.
· Terms are brought up (e.g., CSI) without clear grounding at the beginning of the introduction. The introduction appears, thus, unfocused and should be constrained to a few concepts instead of bringing up new information in every paragraph.
· Lines 76-78: I see here the statement of contribution. Please make sure to state what is the contribution of this paper at the very beginning.
· The literature search involves both mental health and suicidal ideation; however, the whole literature review and the introduction is focused on suicidal attempt/ideation/lived experiences and much less on mental health issues. Try to narrow down what is the research objective. Mental health includes such a broad spectrum of disorders. Can the authors review all these literature sources? The literature search seems a little unfocused.
· Please review in depth extant studies on disclosure and help-seeking relevant to this topic.
· Line 127: what do the authors mean by ‘working in lived experience roles’?
· The inclusion/exclusion criteria are too vague. Why and how was the benchmark of >50% of the sample not working in lived experience roles selected?
· Did the authors include community or clinical samples or both? What are the implications of this?
· Do the authors have interrater reliability of the coding and selection decisions?
· Please clarify to what extent the current study is sufficiently novel compared to previous related systematic reviews such as: Hastuti, R., & Timming, A. R. (2021). An inter-disciplinary review of the literature on mental illness disclosure in the workplace: implications for human resource management. The International Journal of Human Resource Management, 32(15), 3302–3338; Zamir, A., Tickle, A., & Sabin‐Farrell, R. (2022). A systematic review of the evidence relating to disclosure of psychological distress by mental health professionals within the workplace. Journal of Clinical Psychology, 78(9), 1712-1738.
· Lines 411-418: Please clarify what is the new contribution of this study above and beyond what previous empirical works have found. To help the reader understand the novelty, please make sure to review what previous systematic reviews have indicated.
Author Response
Dear Expert Reviewer,
We greatly appreciate your feedback regarding our paper “Disclosure of a mental health problems or suicidality at work: A systematic review’ and ongoing consideration of it for publication in IJERPH.
Attached is a file containing our responses to your advice.
Kind regards,
Authors

Round 2
Reviewer 2 Report
The authors have done a good job in addressing my comments. Their responses are convincing and have revised the manuscript significantly. I do not have any further comments.